clinical practice guidelines; depression; postpartum; postnatal; systematic review

**Corresponding author:**
Aliya Durrani;
Email: a.durrani@keele.ac.uk

# Management of postnatal depression: A systematic review of clinical practice guidelines

Aliya Durrani[1] 🔾, Nishani Fonseka[1], Mirah Rauf Sethi[2], Huma Mughal[1], Zohaib Khan[3], Tom Kingstone[1], Ram Bajpai[1] and Saeed Farooq[1]

[1]School of Medicine, Keele University, Staffordshire, UK; [2]Department of Applied Psychology, Riphah International University, Islamabad, Pakistan and [3]Office of Research Innovation and Commercialization, Khyber Medical University, Peshawar, Pakistan

## Abstract

Postnatal depression (PND) is the most prevalent mental health disorder during the postpartum period. Evidence suggests that clinical practice guidelines (CPGs) can improve the mental well-being of women affected by PND. This study aimed to identify the CPGs available globally for the management of PND and to summarize their recommendations. A comprehensive search was performed across five electronic databases (MEDLINE, PsycINFO, CINAHL, TRIP, and Epistemonikos) and four guideline-specific websites (GIN, SIGN, NICE, and WHO) to identify the English language CPGs published between 2012 and 2023. The general characteristics of the CPGs, as well as the reported pharmacological and non-pharmacological recommendations, were extracted. The AGREE-II instrument was used to assess the methodological quality. Nineteen CPGs were included in the review, with only one from a low and middle-income country (Lebanon). Cognitive-behavioral therapy (CBT) was the most frequently recommended psychological therapy. Pharmacological interventions were included by 17 CPGs, predominantly Selective Serotonin Reuptake Inhibitors (SSRIs). Only three CPGs incorporated Patient and Public Involvement and Engagement (PPIE) in the form of an advisory group. Seven CPGs matched the criteria for adequate methodological quality by achieving an overall score of ≥70%. The findings highlight limited methodological quality and underrepresentation of LMICs, which may lead to disparities in the management of PND and undermine equitable mental health care.

## Impact statement

This systematic review presents an overview of existing CPGs available globally for the management of postnatal depression. Evaluating the scope and methodological quality of these CPGs, this review identifies the best available interventions worldwide, as well as critical gaps in these CPGs, thus offering insights for the guideline development group to produce more robust and evidence-informed CPGs with enhanced applicability. This review also highlights a lack of evidence-based CPGs in most of the LMICs, signaling the necessity for context-specific adaptation. The implications of this review are profound for healthcare professionals, as the viewpoint gained can inform the development of more evidence-based CPGs for PND, thus transforming the quality of maternal mental health. Furthermore, this review will serve as a foundation for the adaptation and integration of adapted CPGs within different health care settings in LMICs, to contribute to better mental health outcomes for women affected by depression in the postnatal period.

## Introduction

Postnatal depression (PND) is a well-known pregnancy complication and the second most common cause of disability, after HIV/AIDS. An estimated 10–15% of women worldwide are thought to be affected by PND (Cheng et al., 2023). It is more prevalent in low and middle-income countries (LMICs), where approximately 20–40% of new mothers are affected by PND (Daliri et al., 2023) compared to high-income nations (HICs), where the prevalence ranges from 6.5 to 12.9% (Wildali et al., 2024). PND has serious consequences for the mother, partner and the infant. Women experiencing PND are at higher risk of developing other mental health issues like anxiety disorder, obsessive-compulsive disorder, bipolar disorder and postnatal psychosis. It has also been linked to higher rates of male partner depression and relationship conflict, which not only compromise the quality of life of both partners but may also impact children's developmental outcomes, including behavioral disturbances, cognitive performance deficits and attachment insecurities in the early years due to maladaptive parenting (Stewart and Vigod,

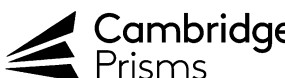



2019). PND may also account for 20% of suicidal deaths in the postnatal period (Lee et al., 2022).

The key barriers to treatment for PND are stigma, low mental health literacy, limited access to mental health care, cultural factors, lack of social support and financial constraints. Additionally, PND often goes undetected and underdiagnosed, as routine screening is not commonly conducted. As a possible solution, several countries have developed clinical practice guidelines (CPGs) to overcome these challenges (Place et al., 2016; Poreddi et al., 2021). CPGs are systematically developed, based on evidence from a rigorous systematic review, and report a set of recommendations on different patient care options (Keating et al., 2017). In addition to their usefulness in clinical settings, CPGs may also impact policy to ensure consistency and uniformity of care delivered in various healthcare settings. Literature suggests that evidence-based guidelines have the potential to improve the care of mothers and newborns and minimize their short- and long-term morbidity (Abdoli Najmi et al., 2023).

As CPGs have the potential to reduce variability in practice, it is important to review the CPGs to summarize the recommendations for PND and to assess the quality of evidence underlying these recommendations. CPGs for the management of PND are severely limited in LMICs, which account for more than half of the global burden of PND. Previous systematic reviews of CPGs have primarily focused on the treatment of peripartum depression, only with antidepressants (Santos et al., 2012) (Molenaar et al., 2018) (Kittel-Schneider et al., 2022). A recent review of CPGs aimed to include only European CPGs containing both pharmacological and non-pharmacological recommendations for peripartum depression (Motrico et al., 2022). To the best of our knowledge, a comprehensive systematic review of the global CPGs for PND, including both pharmacological and non-pharmacological recommendations, is currently lacking in the literature. Therefore, we aimed to review the CPGs available for the management of PND, to collect and collate the pharmacological and non-pharmacological recommendations reported by them. Specifically, this review will address the following questions:

- What are the current CPGs available globally for managing PND?
- What are the specific recommendations reported by these CPGS for the management of PND?
- What is the quality of recommendations and the CPGs?

## Methods

This systematic review was conducted and reported in accordance with the guidelines of the Preferred Reporting Items for Systematic Reviews and Meta-Analyses (PRISMA) (Supplementary Appendix 1). *A priori* protocol was established and registered in the Open Science Framework registry (registration number 10.17605/OSF.IO/9TAYW). The full text of protocol is available at https://archive.org/details/osf-registrations-9tayw-v1

### Data sources and searches

Three electronic databases were searched: MEDLINE, PsycINFO and CINAHL. The search strategy was initially developed in MEDLINE through consultation with an information specialist, and a preliminary search was conducted to refine the search terms. This search strategy was then adapted to PsycINFO and CINAHL. The search also included two guidelines-specific databases, TRIP Database and Epistemonikos. Since many CPGs are not published in journals, an extensive search was also conducted on specific websites of the clinical practice guidelines, including the Guidelines International Network (GIN), World Health Organization (WHO), Scottish Intercollegiate Guidelines Network (SIGN) and National Institute for Health and Clinical Excellence (NICE).

The detailed search strategy and search terms used in the review are given in the supplementary appendix 2 and include both the keywords and Medical Subject Headings related to clinical practice guidelines and postnatal depression. The search strategy was last updated in October 2024. Additionally, hand-searching of the reference lists of included CPGs was carried out to identify other potentially relevant literature. In addition, Google search was also included. The electronic searches of both published and gray literature were performed from 2012 to 2023.

### Eligibility criteria

#### Inclusion criteria

This review focused on published CPGs for the management of PND. The inclusion criteria were (1) CPGs published by a recognized national or international association or professionals and developed and endorsed by an expert group; (2) published in the English language; (3) in full-text format; (4) published from 2012 until 2023, this period is chosen to identify the most recent CPGs.

The exclusion criteria were: (1) Fact sheets and guidelines intended for patients and other end users but not for healthcare professionals, as these lacked the rigor of development or information on stakeholder involvement; (2) We specifically focused on PND and excluded the recommendations and CPGs for the other postnatal psychiatric disorders such as postpartum psychosis; (3) Consensus Statements as the evidence is less extensive here and they are primarily based on expert opinion.

### Selection of CPGs

Results from all the searches were imported into EndNote (reference management software, Clarivate Analytics. Available at https://endnote.com/) and duplicates were removed. The library was then transferred to Rayyan (available at: https://new.rayyan.ai/) to manage the screening process. In the first stage, two reviewers (A.D. and M.R.S.) independently screened titles and abstracts for relevance against eligibility criteria. In the second stage, full texts of the selected CPGs were retrieved and screened for inclusion. Reasons for exclusion at the full-text stage were recorded. Cases of between-reviewer disagreement were resolved through discussion or, where necessary, referral to a third reviewer (R.B.). In addition to the above two stages of screening and inclusion of the final set of CPGs, a randomly selected sample (about 20% of the selected studies) was further reviewed by a research team member (S.F.) to ensure consistency and validation of the selection process.

### Data extraction

The data were extracted independently from each of the CPGs by 2 reviewers (A.D. and H.M.) using a data extraction sheet. It was then independently checked for completion, accuracy and consistency by a third reviewer (R.B.), with any disagreement resolved through discussion. Based on the research questions, the following characteristics were recorded; (1) General characteristics of the guidelines; Title/ author or Institution/Country/ Publication date/ Target users/method of reviewing evidence (2) pharmacological

interventions: recommended medication/dose/duration/method of administration/adverse events (*e.g.,* relapse in mental illness, toxicity, side effects)/advice on use of concomitant medication for PND (3) non-pharmacological treatment; type, components, follow up period. The data extraction sheet is available in the supplementary appendix 3.

### Quality appraisal

The Appraisal of Guidelines, Research, and Evaluation version II (AGREE II) was used to critically appraise the CPGs. AGREE II is an accepted and validated tool for assessing the methodological quality of CPGs internationally. It consists of a 23-item checklist over six domains, and each item is graded on a 7-point Likert scale from 1 (strongly disagree) to 7 (strongly agree). The domains are (i) Scope and purpose of the guideline; (ii) Stakeholder involvement in the development of the guidelines; (iii) Rigor of development and formulation of the recommendations within the guideline; (iv) Clarity of presentation of the guideline; (v) Applicability of the guideline; (vi) editorial independence in the formulation of recommendations within the guideline. In addition, two final items ask appraisers to give an overall judgment based on the 23 items.

Quality assessment was conducted online using the web-based platform My AGREE PLUS (https://www.agreetrust.org/my-agree/) based on information provided in the user's manual. Each item is presented for scoring alongside detailed guidance on how to score the item, including where to find relevant information and what to consider when deciding on the score for each item.

Two reviewers (A.D. and N.F.) independently appraised each CPG. When statements were unclear or ambiguous, guidance from a third reviewer (S.F.) was sought to reach a consensus. As no threshold is specified for determining the high or low quality of the CPGs, the user manual suggests that users decide according to their specific context. Based on the examples given in the AGREE II

user's manual and a previous systematic review (Motrico et al., 2022) A CPG was considered of adequate quality if its overall score was at least 70%. The same cut-off point was used for each of AGREE II's domains.

## Results

### Selection of CPGs

The search strategy identified 1,096 citations, of which 71 CPGs were selected for full-text review. After the exclusion process, 19 CPGs were ultimately included in this review. The PRISMA flow diagram details the selection process, including searches, screenings, reasons for exclusion and selection outcomes (see Figure 1).

### Characteristics of CPGs

An overview of the characteristics of the CPGs is presented in Table 1. The 19 included CPGs originated from Australia ($n = 6$), USA ($n = 3$), UK ($n = 3$), Scotland ($n = 2$), Singapore ($n = 1$), Canada ($n = 1$) and Europe ($n = 1$). Only one originated from LMICs: Lebanon ($n = 1$), and one was published by WHO. More than half of the CPGs were aimed at perinatal and maternal mental health ($n = 15$, 79%), and four addressed perinatal depression specifically (21%).

### Recommendations for the treatment of postnatal depression

The core recommendations are summarized in Table 3.

#### Non-pharmacological recommendations
All the CPGs recommended psychological therapy, with CBT being recommended by all as 1st line option for the management of mild-to-moderate depression in the postnatal period, with 8–12 consecutive

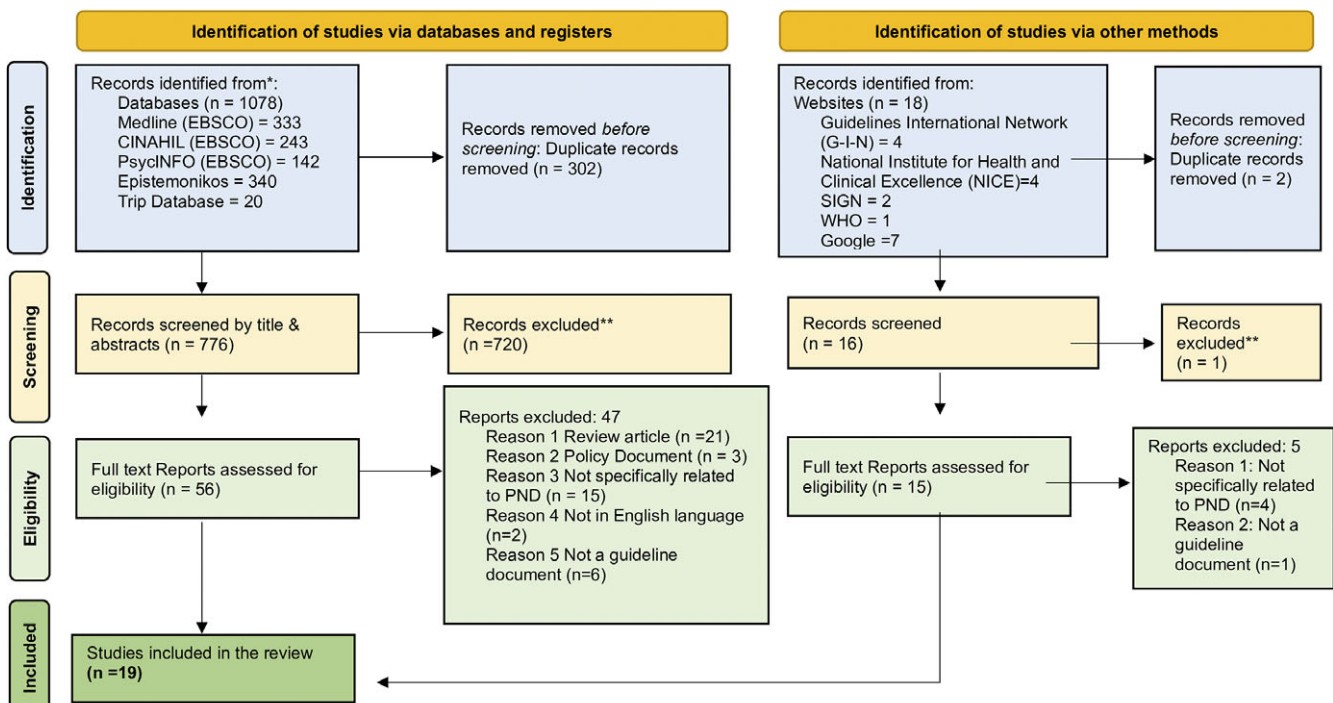

**Figure 1.** PRISMA flow diagram.

**Table 1.** Characteristics of the included clinical practice guidelines

| Sno. | Title | CPG Organization/society/institution and country | Publication year | Target users |
|---|---|---|---|---|
| 1 | Treatment and management of mental health conditions during pregnancy and postpartum | American College of Obstetricians and Gynecologists (ACOG) Clinical Practice Guideline, USA | 2023 | Clinicians in providing obstetric and gynecologic care |
| 2 | Antenatal and postnatal mental health: clinical management and service guidance | National Institute for Health & Excellence (NICE) UK | 2018 | Healthcare professionals, Commissioners, Social Services, Voluntary and private sectors |
| 3 | Mental health care in the perinatal period: Australian clinical practice guideline | Centre of Perinatal Excellence (COPE), Australia | 2023 | All health professionals who care for women and families during the perinatal period |
| 4 | Best practice guidelines for mental health disorders in the perinatal period | Reproductive mental health program, Mental Health and substance use services and perinatal Services in British Columbia, Canada | 2014 | Healthcare clinicians who care for women during the reproductive years |
| 5 | SIGN169: perinatal mental health conditions: a national clinical guideline | Healthcare Improvement Scotland (HIS) SIGN, Scotland | 2023 | Family nurse practitioners, general practitioners (GPs), health visitors, midwives, nursery nurses, obstetricians, occupational therapists, pharmacists, psychiatrists, psychologists, psychotherapists, social workers |
| 6 | Perinatal and infant mental health model of care – a framework | Government of Western Australia Department of Health, Australia | 2016 | GP, obstetrician, psychiatrist, neonatologist, midwife, psychologist and any other involved professionals collaborating actively in developing a perinatal mental health plan |
| 7 | British Association for Psychopharmacology consensus guidance on the use of psychotropic medication preconception, in pregnancy and postpartum | British Association for Psychopharmacology, UK | 2017 | Professionals involved in evidence-based practice |
| 8 | Perinatal mental health guidelines on depression and anxiety | College of Obstetricians & Gynecologists, Singapore | 2023 | GPs, Family Medicine/ Primary Health Physicians, gynecologists and obstetricians, Pediatricians, Nurses, mental health professionals |
| 9 | Perinatal depression screening, diagnosis and treatment guidelines | Kaiser Foundation Health Plan of Washington, USA | 2021 | Healthcare professionals in the primary settings and primary care social workers |
| 10 | Mental health care in the perinatal period | Centre of Perinatal Excellence (COPE), Australia | 2017 | Health professionals caring for women and families during the perinatal period. This includes but is not limited to midwives, GPs, obstetricians, neonatologists, pediatricians, maternal and child health nurses, pediatric nurses, Aboriginal and Torres Strait Islander health workers, allied health professionals, mental health practitioners (psychologists, psychiatrists, mental health nurses, perinatal and infant mental health professionals), consumers and carers and those working with families in the community (*e.g.* social workers, child protection agencies), hospitals and legal systems. |
| 11 | Perinatal mental health and psychosocial assessment: practice resource manual for Victorian maternal and child health nurses | Department of Education and Early Childhood Development by the Parent-Infant Research Institute (PIRI), Austin Health and Mercy Women's Hospital Perinatal Mental Health Service, state government of Victoria, Australia | 2013 | Maternal and child health nurses |
| 12 | Assessment and interventions for perinatal depression | Registered Nurse Association, Ontario, Canada | 2018 | Nurses, the interprofessional team (including, but not limited to, physicians, midwives, social workers, lactation consultants and psychologists), educators, policymakers and persons and their families to improve outcomes. |

**Table 1.** (*Continued*)

| Sno. | Title | CPG Organization/society/institution and country | Publication year | Target users |
|---|---|---|---|---|
| 13 | The perinatal mental health care pathways. full implementation guidance | National Collaborating Centre for Mental Health. London UK | 2018 | Clinical Commissioning Group (CCG), mental health commissioners and providers of perinatal mental health services, statutory and non-statutory social care providers and local authorities, working collaboratively with women who use perinatal mental health services and their families and carers. |
| 14 | Evidence-based clinical practice guidelines for prevention, screening and treatment of peripartum depression | This publication is based upon work from the COST Action Research Innovation and Sustainable Pan-European Network in Peripartum Depression Disorder (Riseup-PPD), CA18138, supported by COST (European Cooperation in Science and Technology), Europe | 2023 | Mental health professionals (MHP) (such as psychiatrists, psychologists, counselors, psychosomatic medicine practitioners and other MHPs, including midwives, obstetricians/gynecologists, pediatricians, nurses, general practitioners, social workers, pharmacists and others who play key roles in developing and implementing interventions for the prevention, screening, or treatment of PND. Further targets for these guidelines also include politicians, economists, policymakers and non-profit organizations, including patient organizations, who may be involved in decision-making about funding, developing and implementing interventions for preventing, screening, or treating PND |
| 15 | Perinatal mental health and psychosocial assessment: practice resource manual for Victorian maternal and child health nurses | Department of Health and Human Services. Originally published by the Department of Education and Training. State of Victoria, Australia | 2019 | Maternal and child health nurses |
| 16 | Detection and management of mood disorders in the maternity setting: The Australian Clinical Practice Guidelines | Perinatal and Women's Mental Health Unit, St John of God Hospital, Australia | 2013 | Range of clinicians — midwives, early childhood nurses, general practitioners, obstetricians, mental health professionals — caring for women and their families across the perinatal period and covering the detection and management of perinatal mood disorders, anxiety and psychosis |
| 17 | SIGN 127 • Management of perinatal mood disorder | SIGH, Scotland | 2012 | Midwives, health visitors, general practitioners, pharmacists, nurses, psychiatrists, obstetricians, neonatologists, pediatricians, clinical psychologists, social workers, public health physicians, users of services and all other professionals caring for women and their families |
| 18 | Maternal mental health guidelines for healthcare providers | National Mental Health Program (NMHP) at the Ministry of Public Health (MOPH), in collaboration with the United Nations Population Fund (UNFPA) in Lebanon | 2021 | Obstetric gynecologists, midwives, nurses, pediatricians |
| 19 | Guide for integration of perinatal mental health in maternal and child health services | WHO | 2022 | Clinical managers and health service administrators at hospitals, district and primary health facilities and in non-governmental organizations (NGOs) and community-based organizations that provide Mother Child Health services and allied health professionals |

sessions. This recommendation is supported by high-quality evidence (indicating that it is based on systematic reviews and that the estimated effect is close to the true effect). In addition to the face-to-face delivery format, four CPGs (22%) also considered online approaches for CBT. Thirteen CPGs (68%) recommended Interpersonal Therapy (IPT) as an evidence-based intervention for PND (high-quality evidence). Additionally, psychodynamic therapy, mother-infant psychotherapy and couple therapy were also recommended by three CPGs (16%). Twelve CPGs (63%) recommended that these psychotherapies be conducted by health professionals such as psychiatrists, psychologists, general practitioners, occupational therapists, social workers and nurses trained in these therapies. This recommendation is supported by low-quality evidence (indicating that it is based on observational studies, and there is a substantial difference between the true and estimated effect).

### Pharmacological recommendations

Pharmacological recommendations were included in 17 CPGs (89%), with SSRIs as the 1st-line pharmacological option,

recommended by 11 CPGs (58%) for the management of moderate to severe PND (high-quality evidence). Among the SSRIs, paroxetine, sertraline, citalopram and escitalopram are recommended by 58% CPGs (*n* = 11). They are safer drugs during breastfeeding as they have the lowest degree of passage into breastmilk; however, women should be advised to take them with food, and they generally take 7–8 days to work. Long-acting antidepressants like Fluoxetine are not recommended by 37% (*n* = 7) CPGs, due to the potential risk of accumulation in breastmilk. According to the CPG from the American College of Obstetricians and Gynecologists (ACOG) and the Best Practice Guidelines for Mental Health Disorders in the Perinatal Period, British Columbia, the therapeutic doses for the recommended SSRIs are listed in Table 2.

In addition to SSRIs, 4 CPGs (21%) also recommend Tricyclic Antidepressants (TCAs) and Serotonin-Norepinephrine Reuptake Inhibitors (SNRIs). ACOG and the European (RISE-UP PPD) CPGs recommended the use of the new drug Brexanolone, a formulation of allopregnanolone (a neurosteroid), for moderate to severe depression in the postnatal period. This recommendation is supported by moderate quality evidence (indicating that it is based on randomized controlled trials with some limitations, and the estimated effect is likely to be close to the true effect). This drug requires an IV infusion and has a faster onset of action (symptom reduction in 1–2 days) as compared to the oral antidepressants. Evidence is scarce on the safety of brexanolone exposure *via* breastmilk on the infant.

Four CPGs recommended Electro Convulsive Therapy (ECT) for women with severe depression during the postnatal period, when there is a failure of response to one or more trials of antidepressants with adequate dose and duration. According to these CPGs, it is a first-line treatment for postnatal women with severe depression, especially when there is a higher risk of suicide or at a high level of distress, when the food or fluid intake is poor (low-quality evidence).

### *Other recommendations*
- **Prevention, psychosocial assessment and screening tools**

Recommendations on the prevention of PND were included by 6 CPGs (32%) with low-quality evidence. Nine CPGs (47%) provide recommendations to undertake the psychosocial assessment in conjunction with a screening tool for depression in the postnatal period (low-quality evidence). Thirteen CPGs (68%) provide recommendations for the screening of PND using the EPDS tool (high-quality evidence). However, other standardized scales, such as the PHQ-9 tool, were recommended by 4 CPGs (21%) with low-quality evidence, and the Hooley questions by 2 CPGs (11%), with very low-quality evidence (Based on case reporting and expert opinion, and the confidence in the estimated effect is very low).

**Table 3.** Summary of core recommendations

| S.no. | Pharmacological recommendations | No. of CPGs |
|---|---|---|
| 1 | Provide counseling on the risks and benefits of starting pharmacological treatment, including the potential consequences of untreated depression. | *n* = 14 |
| 2 | Consider the use of Selective Serotonin Reuptake Inhibitors as first-line treatment for moderate-to-severe depression in postnatal women. | *n* = 11 |
| 3 | Pharmacotherapy should be individualized based on prior response to the therapy. | *n* = 9 |
| 4 | Provide support for women in their decision about breastfeeding and be aware that antidepressant use is not an absolute contraindication to breastfeeding. | *n* = 8 |
| 5 | Before prescribing antidepressants to women who are breastfeeding, consider the infant's health and gestational age at birth. | *n* = 7 |
| 6 | Use the lowest effective dose needed to achieve the therapeutic response. | *n* = 4 |
| 7 | Minimize switching of medications and use a single drug, in preference to two or more drugs. | *n* = 4 |
| | **Non-pharmacological recommendations** | |
| 1 | Recommend individual structured psychological intervention (cognitive behavioral therapy) to women with mild to moderate depression in the postnatal period. | *n* = 19 |
| 2 | Recommend Interpersonal therapy to women with mild to moderate depression in the postnatal period. | *n* = 13 |
| 3 | Offer facilitated self-help to women with depressive symptoms in the postnatal period. | *n* = 5 |

- **Psychoeducation and lifestyle advice**

Structured psychoeducation was recommended by 10 CPGs (53%) to improve depressive symptoms during the postnatal period (high-quality evidence). Moreover, lifestyle advice in the form of healthy eating, adequate physical activity and regular sleep patterns was recommended by 8 CPGs (42%) with low-quality evidence.

- **Referral and care pathway**

Seven CPGs (37%) included recommendations (low-quality evidence) about the referral pathway for women at risk of or experiencing PND, or those not responding to the treatment. These CPGs recommend that women should be seen by a midwife in the first six weeks postnatally and then transition to a maternal and child health care nurse, who provides ongoing support and care. If there is probable depression, the woman will be referred to a General Practitioner who will diagnose and develop a management plan.

**Table 2.** First-line pharmacological treatment for postnatal depression

| Medication | Sertraline | Paroxetine | Citalopram | Escitalopram |
|---|---|---|---|---|
| Starting dose | 50 mg × 7 days qAM (if sedating change to qHS) | 10–20 mg qAM | 10–20 mg qAM | 5–10 mg qAM |
| Increase after 7 days | ↑ to 100 mg | ↑ by 10 mg | ↑ by 10 mg | ↑ by 10 mg |
| Reassess monthly | ↑ by 50 mg | | | |
| Therapeutic range | 50–200 mg/day | 20–60 mg/day | 20–40 mg/day | 20–40 mg/day |

If the symptoms are more serious, a direct referral to a psychiatrist is recommended, who may be responsible for developing a mental health treatment plan for this woman.

### Development and quality appraisal of CPGs

Eleven CPGs (58%) used the guideline development group, whereas 42% utilized the expert collaboration for the development of CPGs. For the determination of the strength of recommendation, "GRADE Criteria" were employed by 9 CPGs (47%), though only one used the AGREE II checklist. The systematic review was used as the level of evidence in 68% ($n$ = 13) of CPGs. Five CPGs (26%) included a search strategy in the appendix, and only 3CPGs (16%) incorporated the views and preferences of the patient and the public in their development.

The AGREE-II domain score for each CPG is detailed in Table 4. The overall assessment score for the quality appraisal of CPGs ranged from 35.29% to 87.14% (mean = 62.44%). Seven out of the nineteen CPGs (37%) were considered highly rated and achieved an overall score of ≥70% (highlighted in Table 3). Over half of the CPGs ($n$ = 16, 84%) demonstrated a high-rated score for the fourth domain, *i.e.*, "Clarity and Presentation" (mean 85.96%, range = 58.33% to 100%), except for three CPGs (Serial no of CPGs: 7, 11 and 13). The main factors responsible for gaining a high score in this domain were that the recommendations were specific, unambiguous and easily identifiable. Furthermore, most of the CPGs ($n$ = 17, 89%) indicated a low-rated score for the fifth domain, *i.e.*, "Applicability" (mean = 38.04%, range = 10.41% to 79.16%), apart from only two CPGs (Serial no. of CPGs 2 and 3). The low scores in this domain were due to a lack of clear description of facilitators and barriers to the application of clinical recommendations and failure to detail the cost-related information.

## Discussion

We used a comprehensive search strategy to map all the evidence and found 19 CPGs, with only one from LMICs. The main findings of our review are a) all the CPGs recommended non-pharmacological interventions (CBT and IPT) as the first-line intervention for PND, and 17 provided recommendations on pharmacological treatment. b) Of the 19 CPGs, 13 provided recommendations on screening of PND, 10 for psychoeducation, 9 for the psychosocial assessment, 8 for lifestyle advice, 7 for the referral pathways and 6 for prevention interventions. c) Data extraction showed considerable variation in the CPG development process. d) Seven CPGS demonstrated adequate quality with a score of 70% and above. However, a gap observed in the latest available evidence is the lack of information regarding emerging issues and questions such as drug monitoring, dose adjustment and antidepressant switching.

In our review, non-pharmacological interventions (CBT and IPT) have been recommended for the management of mild to moderate depression in postnatal women. They are highly effective interventions in reducing depressive symptoms during the postnatal period, as supported by high-quality evidence, with their efficacy further supported by the US Preventive Services Task Force's systematic review and other reviews as well (Branquinho et al., 2021; Motrico et al., 2023; O'Connor et al., 2019). Our review showed that CBT can be delivered face-to-face as well as through an online delivery format. Previous reviews also highlighted that psychological interventions provided at a distance *via* online platforms can effectively improve depressive symptoms among postnatal women (Lee et al., 2016; Nillni et al., 2018). A consensus report from the COST Action Rise up-PPD suggests that both the individual and group therapy formats grounded in CBT or IPT are efficacious in PND (Fonseca, Lambregtse-van den Berg and Rodriguez-Muñoz, 2020), a finding supported by our review as well.

In our review, SSRIs are recommended for moderate to severe depression during the postnatal period, with a focus on conducting risk–benefit analysis regarding breastfeeding and infants' health. Among the SSRIs, the most prescribed medications are sertraline, citalopram and escitalopram due to their minimized risk during breastfeeding; these findings are in alignment with other reviews (Kittel-Schneider et al., 2022; Molenaar et al., 2018). In our review, CPGs recommended the importance of the following key factors in treatment decision-making: psychiatric history, current psychiatric symptoms, prior response to the drug, monotherapy and use of the lowest effective dose.

ECT for severe depressive symptoms in the postnatal period is recommended by four CPGs; however, there is uncertainty regarding the use of ECT, which is consistent with the previous reviews. According to the previous reviews, the use of ECT in severe cases of PND showed positive results due to its more rapid onset of action as compared to standard antidepressants. However, further investigation is needed to evaluate the risk–benefit ratio and effects of ECT on both the mother and fetus, as the current evidence is based on moderate-quality case reports (Pacheco et al., 2021; Gressier et al., 2015; Rundgren et al., 2018).

The EPDS screening tool is recommended by this review with high-quality evidence for the detection of depression among postpartum women, a tool also supported by prior reviews as well (Park and Kim, 2023; Daskalakis et al., 2024). Our review highlights the lack of quality evidence in the psychosocial assessment of postnatal women, along with the screening. Literature supports the role of psychosocial factors like stressful life events, poor quality of the relationship and lack of support from husband/partner and family in PND, demonstrating the need to understand these factors during the treatment of PND (Yim et al., 2015; Tenaw et al., 2024). Furthermore, a gap in evidence is observed regarding recommendations on lifestyle factors such as exercise, a healthy diet and quality sleep. Previous reviews suggest that engaging in exercise or physical activity and maintaining a healthy diet can significantly reduce symptoms of PND (Brunson et al., 2024) (Ghaedrahmati and Alipour, 2024).

PPIE is internationally recognized as a key element of evidence-based CPGs. Involvement of the patients in the development of CPGs improves the quality of their care, particularly when considering preference-sensitive decisions, such as the treatment options for the PND (Björkqvist et al., 2021; Pfisterer-Heise et al., 2025). However, this review reflects a lack of PPIE in most of the existing CPGs. Despite the international recommendation supporting the inclusion of patients in the production of CPGs, there is a lack of clear methodology and insufficient data regarding the optimum stage for the involvement of patients.

This review highlights that only seven CPGs met the criteria for adequate quality. The three domains, *i.e.*, rigor of development, applicability and editorial independence, received the lowest score across all the CPGs, similar findings are reported by the previous reviews as well (Motrico et al., 2022; Molenaar et al., 2018). The guideline developers should strive to include the update procedure to maintain alignment with the most recent evidence in clinical practice. To enhance the rigor of the development domain, the CPGs should provide the details of search terms used for obtaining quality evidence. Furthermore, the CPGs should address the financial implications of guideline implementation and the auditing criteria to strengthen the applicability domain.

**Table 4.** Quality appraisal of CPGs using the AGREE-II instrument

| S.No. | Name of CPG | Domain 1 Scope and purpose | Domain 2 Stakeholder involvement | Domain 3 Rigor of development | Domain 4 Clarity of presentation | Domain 5 Applicability | Domain 6 Editorial independence | Overall assessment |
|---|---|---|---|---|---|---|---|---|
| 1 | Treatment and management of mental health conditions during pregnancy and postpartum, ACOG | **77.77%** | 69.44% | 64.58% | **86.11%** | 27.08% | 62.50% | 64.58% |
| 2 | Antenatal and post-natal mental health: clinical management and service guidance, NICE UK | **97.22%** | **88.88%** | **83.33%** | **97.22%** | **77.08%** | **79.16%** | **87.14%** |
| 3 | Mental Health Care in the Perinatal Period 2023 Australia | **86.11%** | **91.66%** | **73.95%** | **91.66%** | **79.16%** | **75%** | **82.92%** |
| 4 | Best practice guidelines for mental health disorders in the perinatal period, BC, USA | **86.11%** | 72.22% | 48.95% | **91.66%** | 27.08% | 45.83% | 61.97% |
| 5 | SIGN169: Perinatal mental health conditions, Scotland | **80.55%** | **88.88%** | **77.08%** | **97.22%** | **47.91%** | **58.33%** | **74.99%** |
| 6 | Perinatal and infant mental health model of care – a framework, Australia | **72.22%** | 75% | 18.75 | **77.77%** | 43.75% | 37.50% | 54.16% |
| 7 | British Association for Psychopharmacology consensus guidance on the use of psychotropic medication preconception, in pregnancy and postpartum 2017, UK | **72.22%** | 69.44% | 37.50% | 63.88% | 10.41% | 50.00% | 50.57% |
| 8 | Perinatal mental health guidelines on depression and anxiety, Singapore | **80.55%** | 69.44% | 28.12% | **77.77%** | 25% | 25% | 50.98% |
| 9 | Perinatal depression screening, diagnosis and treatment guideline, USA | 66.66% | 47.22% | 32.29% | **83.33%** | 12.5% | 8.33% | 41.72% |
| 10 | Mental Health Care in the Perinatal Period 2017 Australia | **83.33%** | **80.55%** | **92.70%** | **97.22%** | 66.66% | **100%** | **86.74%** |
| 11 | Perinatal Mental Health and Psychosocial Assessment 2013, Australia | 66.66% | 36.11% | 21.87% | 69.44% | 22.91% | 29.16% | 41.02% |
| 12 | Assessment and Interventions for Perinatal Depression, Canada | **91.66%** | **91.66%** | **81.25%** | **94.44%** | 60.41% | **75%** | **82.40%** |
| 13 | The Perinatal Mental Health Care Pathways, NHS UK | **86.11%** | **86.11%** | 15.62% | 58.33% | 35.41% | 12.5% | 49.01% |
| 14 | Evidence-Based Clinical Practice Guidelines for Prevention, Screening And Treatment of Peripartum Depression, Europe (RISE-UP PPD) | **88.88%** | **91.66%** | **67.70%** | **97.22%** | 27.08% | **83.33%** | **75.97%** |
| 15 | Perinatal mental health and psychosocial assessment 2019, Australia | 55.55% | 47.22% | 14.58% | **83.33%** | 25% | 33.33% | 35.29% |
| 16 | Detection and management of mood disorders in the maternity setting: The Australian Clinical Practice Guidelines, Australia | **80.55%** | **80.33%** | 56.25% | **86.11%** | 22.91% | 37.5% | 60.60% |
| 17 | SIGN 127 • Management of perinatal mood disorder, Scotland | 66.66% | **83.33%** | 67.70% | **97.22%** | 27.08% | **83.33%** | **75.97%** |
| 18 | Maternal Mental Health Guidelines for Healthcare Providers, Lebanon | **72.22%** | 50% | 34.37% | **86.11%** | 22.91% | 50% | 52.60% |
| 19 | Guide for integration of perinatal mental health in maternal and child health services, WHO | 69.44% | **77.77%** | 44.79% | **94.44%** | 41.66% | 66.66% | 65.79% |

*Note*: Seven out of the nineteen CPGs (37%) were considered highly rated and achieved an overall score of ≥70% (highlighted in Table 3).

The recommendations and interventions identified in our review are associated with beneficial outcomes in LMICs, as supported by the existing literature. Previous systematic reviews demonstrated that the application of CBT and IPT in low-resource settings can alleviate the symptoms of PND (Nillni et al., 2018). However, they emphasized that these structured psychotherapies can be culturally adapted and delivered by community health workers in these settings, which is referred to as the "task shifting" approach (Gajaria and Ravindran, 2018). In LMICs, the pharmacological treatment for PND most commonly involves the use of SSRIs, particularly sertraline, which aligns with the findings of our review (Azhar et al., 2024). Nevertheless, it has been claimed that access to psychiatric medication in LMICs is limited, which may lead to the least available data on pharmacological interventions for PND in these countries (Gajaria and Ravindran, 2018). In LMICs, ECT is used for treatment-resistant depression; however, unmodified ECT is commonly practiced due to its low cost and limited resource requirement (Daskalakis et al., 2024). Additionally, EPDS is psychometrically valid for use in low-income settings owing to its acceptable sensitivity, specificity and accuracy. It is routinely used here in maternity care due to its affordability and easy administration (Chorwe-Sungani and Chipps, 2017; Fellmeth et al., 2021).

This review highlights that CPGs to guide the management of PND are not available in most of the LMICs. Therefore, the healthcare providers in these countries depend on CPGs developed by international organizations and professional bodies in HICs. However, CPGs established in HICs are difficult to use in the LMIC setting due to major differences in socioeconomics, demographics, culture and pharmacodynamics (Olayemi et al., 2017). According to the literature, local guidelines are more likely to be effective due to the sensitive nature of PND. So, the best course of action is to involve the local stakeholders and experts to adapt the currently available guidelines for PND that will suit the needs and cultural practices of the local community (Olayemi et al., 2017; Johnston et al., 2019).

## Strengths and limitations

To the best of our knowledge, this is the first review to provide a global overview of the currently available CPGs for the management of PND. This review enabled us to identify the up-to-date treatment options available for the current practices across the globe due to their focus on contemporary CPGs. To strengthen the quality of this evidence summary, inclusion–exclusion criteria were developed, quality appraisal of the CPGs was done by using the AGREE checklist, and a clear criterion was set for summarizing the recommendations from all the included CPGs. Independent assessment of the eligibility criteria, data extraction and quality assessment by different reviewers ensures consistency and improves the strength of this evidence synthesis.

Our review was restricted to CPGs published in English; this may lead to excluding other relevant CPGs that may bring a different cultural perspective to this systematic review. Another limitation is the potential for subjective judgment when using the AGREE checklist. However, this limitation was overcome by using the opinions of multiple independent reviewers. Moreover, there is a lack of validated cut-off points published for the AGREE-II checklist; therefore, the reference for the cut-off point was drawn from the previous studies. Finally, this review did not include Clinical Consensus Statements (CCS), expert opinion or the articles as they were not developed according to the CPGs and are based on individual perspectives and practices.

## Implications for research and clinical practice

Future updates should consider the latest evidence and a broader range of stakeholder perspectives. International standards and a uniform procedure should be followed in the development of new CPGs, with a focus on rigor of development, applicability and editorial independence domains to ensure high quality. In addition, Patient public involvement and engagement are significant in the development of the CPGs to incorporate the perspective of PND patients into the recommendations, to make them more patient-centred and practical, and to build trust with health care professionals. Integrating CPGs into clinical practice can potentially improve early detection and prompt treatment. CPGs advocate for a multidisciplinary approach by involving gynecologists and obstetricians, primary care physicians, midwives and mental health professionals to ensure the provision of comprehensive care to new mothers, addressing both their physical and mental needs.

## Conclusion

This review highlights that most of the CPGs belong to HICs, with only one representing the LMICs. CBT and SSRIs are the most recommended interventions across these CPGs. In addition, it emphasizes the concern that the low methodological quality of most of the CPGs could lead to disparities and inequalities in the treatment of PND. So, this review underscores the need for harmonized and up-to-date CPGs supported by quality evidence for the treatment of PND. The overall goal is to empower women and enhance their access to evidence-based, standardized care for the management of PND globally.

**Open peer review.** To view the open peer review materials for this article, please visit http://doi.org/10.1017/gmh.2025.10075.

**Supplementary material.** The supplementary material for this article can be found at http://doi.org/10.1017/gmh.2025.10075.

**Data availability statement.** All data related to the study are available within the article, references, or its supplementary material.

**Author contribution.** AD, NF and SF designed the study protocol with input from TK, RB and ZK. AD designed the literature search with input from SF and NF. AD and MS performed the study selection with input from SF, TK, RB and ZK. AD and HM carried out data extraction, while the quality assessment was conducted by AD and NF with input from SF. AD carried out the analysis and interpretation of the data with input from SF. AD and NF drafted the manuscript. All authors critically revised the manuscript for intellectual content and read and approved the final manuscript.

**Financial support.** This review is a part of Doctoral Studies under the National Institute for Health and Care Research (NIHR) funded project titled "Cognitive Therapy of Depression Treatment in TB Patients (The CONTROL). CONTROL NIHR201773/ PID-200016.

**Competing interests.** The authors have no conflicts of interest to declare.

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
