## [Reviewer Report]

This is a very useful paper. It is very unfortunate that they only reviewed guidelines published in English, because much can be learned from guidelines published elsewhere and we would have a more balanced view.

The paper is easy to read and it will be useful for junior staff, because it explains terms as they appear (for instance, page 13 “high-quality evidence (indicating that it is based on systematic reviews and that the estimated effect is close to the true effect)…” and so on. The reader then gets a complete message in one sentence. This way the detailed work done by the authors can achieve its goal: to shed light on the treatment of this highly prevalent disorder which invoves the health (physical and mental) of the woman and the baby.

The findings are comforting: there seems to be agreement about what is useful and what isn’t.

The paper would probably benefit from some native English speaker specialized edition, but I really look forward to seeing it in print.

---

## [Reviewer Report]

This review provides a global overview of the currently available CPGs for the management of PND. This review is very timely and relevant to the field of perinatal mental health. In general, this manuscript is well-written. However, this manuscript can be improved in the following ways.

1. Inclusion criteria: This manuscript included CPGs published by a recognized national or international association or professionals and developed and endorsed by an expert group. However, it is not clear whether the authors included consensus statements as well.

2. The authors may want to discuss more about the implications of only including CPGs in English. As many LMICs or even HICs may release CPGs or consensus statements in other languages, this manuscript is likely to underestimate the overall number of CPGs in LMICs.

3. In the second paragraph of the Discussion section, the authors mentioned that “our review showed that CBT can be delivered face-to-face as well as through an online delivery format.” However, the authors did not mention this point in the results section.

4. The authors mentioned PPIE in the Discussion section. However, this abbreviation has not been mentioned before. Please clarify this abbreviation in the manuscript.